# Enforcing and Resisting Hindutva: Popular Culture, the COVID-19 Crisis and Fantasy Narratives of Motherhood and Pseudoscience in India

**Catarina Kinnvall** [1,*] **and Amit Singh** [2] 

1   Department of Political Science, Lund University, Box 52, 221 00 Lund, Sweden
2   Centre for Social Studies, University of Coimbra, Colégio de S. Jerónimo, 3000-995 Coimbra, Portugal
*   Correspondence: catarina.kinnvall@svet.lu.se

**Abstract:** This article analyzes how Hindu nationalists employ fantasy narratives to counteract resistance, with a particular focus on narratives of 'motherhood' and 'pseudoscience'. It does so by first introducing a conceptual discussion of the relationship between fantasy narratives, ontological insecurity, gender, and anti-science as a more general interrelationship characterizing pre- and post-COVID-19 far-right societies and leaders, such as India. It then moves on to discuss such fantasy narratives in the case of India by highlighting how this has played out in two cases of Hindu nationalist imaginings: that of popular culture, with a specific focus on the town Varanasi and the film Water (produced in 2000), and that of the COVID-19 pandemic and the emerging crisis and resistance that it has entailed. Extracts of interviews are included to illustrate this resistance.

**Keywords:** resistance; COVID-19; far right; nationalism; gender; pseudoscience; ontological security; India

## Introduction

The COVID-19 pandemic may currently be in its final phase considering the opening up and easing of restrictions in many parts of the world, but its lasting legacies are still to be understood and evaluated in relation to past and present nationalist (and other) practices. Most important is how the pandemic shaped a widely dispersed state of uncertainty and how responses to such uncertainties contained various kinds of memorialized nationalist story-making. Quarantine, lockdowns, and closed borders largely induced a 'new normality' by undermining and unsettling the ordinary routines that create a sense of continuity and provide answers to questions about 'doing, acting, and being', what Giddens (1991) refers to as ontological security. The pandemic thus functioned as a crisis narrative by exposing the fragility of political life, with its intrinsic doubts, anxieties, and uncertainties. In this sense, the political response to the pandemic was closely tied to the conjuring up of secure images of the future, involving some kinds of fantastical predictions and visions concerning what that future may look like, how particular re-memorialized pasts are to be blamed or glorified, and the specific actors responsible for the crisis. As much work on far-right populism has shown, such responses often take on a nativist and gendered dimension and are particularly common among politicians and leaders on the far right, who aim towards narrative closure of what a nation is, who is to be its rightful owners, and who should be excluded.

These fantasy narratives have also converged in many Hindu populist imaginings, in which specific emotions become tied to fantastical visions of the past, present, and future, involving everything from fabricated lies to myths and re-imagined memories of a past glorious order. Traditional and digital media discourses, as well as popular culture, have been crucial for the spread of such sentiments and have allowed for visual symbols, imagery, and mythological tales about imaginary pasts and 'others' to be securitized and

sedimented. As political storytelling, popular culture provides a prominent avenue for both hegemonic discourse and resistance to take form. Either in terms of hegemonic (and in our case gendered nativist) stories with fantasized beginnings, presents, and ends in response to ontological insecurities, or as ruptures to this hegemonic logic with a possibility to resist, disturb, or counter the hegemonic narrative. Through its increasing digitization, political storytelling has enabled access to mass audiences at the same time as it has allowed for fabricated images, fantasies, and myths to take the shape of 'real' events and 'real' historical patterns. In this article, we discuss these myths and re-imagined memories in relation to nationalist fantasy narratives in India and how resistance to such narratives has been met by a hegemonic repressive Hindu, or Hindutva, populist discourse before and during the time of the current Prime Minister Narendra Modi.

As a nationalist narrative, Hindutva thrives on fantasies originating from the Indian past and centers on narratives of 'Indian motherhood', often in relation to fantastical (fictional) images of Muslim (and other minority) others. Hindu nationalism relies on the idea that India, that is Bharat, once a Vishwa guru—the master of the world—is the original home to Hindus. In the nationalist narrative, only Hindus are the authentic natives of the land and Hindu women are sacred goddesses born to serve the family and the nation. Relying on these fantasy narratives, Hindu nationalists tend to portray 'resistance' to this national narrative as 'anti-national' (anti-Bharat) and 'anti-Hindu' (Anand 2011; Juergensmeyer 2000; Kaul 2021; Kinnvall 2006). These fantasy narratives do not exist in a vacuum, however, but have been met by defiance from various civil society movements of the Indian society, such as minority groups, secularists, academics, and human rights activists, manifesting at the local to the national levels and organized by both individuals and numerous organizations. Some of this resistance is ad-hoc, un-coordinated, and limited to the elites, whereas some has developed into major political movements with large popular support. The response from Hindutva forces to such resistance has been to reject the claims of these groups or individuals through a discourse of cultural fantasy narratives.

This article hence analyzes how Hindu nationalists employ fantasy narratives to counteract resistance and construct an illusory form of ontological security for those identifying as Hindus. This entails a particular focus on how the narrative of 'motherhood' has defined Hindu nationalist fantasies over time and how this narrative became linked to narratives of 'pseudoscience' during the COVID-19 pandemic. It does so by first introducing a conceptual discussion of the relationship between fantasy narratives, insecurity, gender, and pseudo/anti-science as a more general interrelation characterizing far-right societies and leaders, such as India. It then moves on to discuss such fantasy narratives in the case of India by highlighting how Hindu nationalist imaginings have played out in two cases: that of popular culture, with a specific focus on the film Water (produced in 2000), and that of the COVID-19 pandemic and its emerging crisis and resistance. These events are more than 20 years apart and were not chosen for comparative reasons, but to explore and illustrate how Hindu nationalists have consistently used fantasy narratives of 'motherhood' as ontological security-seeking practices, and how resistance to such narratives has continued to be met by repression and violence. Narratives of the anti-Water Hindutva movement were about reaffirming nostalgia in terms of the sacredness of Hindu women in ancient India and their imposition in the modern context. The movement was also about asserting the superiority of Hindu culture and ancient Indian mythical knowledge. Both have re-emerged in the recent pandemic context through a privileging of a pseudoscience that takes its point of departure in narrative fantasies of Vedic science and motherhood. Hence, the article shows how continuity in fantasy narratives can work as ontological security-seeking practices over time that reassert dominance and hegemony in the face of resistance. Extracts of interviews are included to illustrate both the persistence of and the resistance to such fantasy narratives.[1]

Hence, the next section provides a theoretical overview of ontological insecurities and gendered fantasy narratives with a particular focus on the far right and anti/pseudoscience. We then move on to show how such gendered fantasy narratives, in particular narratives of

'motherhood', have been employed and resisted in relation to popular culture and the film Water. Following this, we highlight some cases of resistance against the Hindu nationalist government of Narendra Modi before and during the COVID-19 pandemic. Finally, we discuss gendered fantasy narratives of motherhood in relation to how 'pseudoscience' as Vedic science was employed by Hindu nationalists during the pandemic in face of such resistance.

**Ontological Insecurities and Gendered Fantasy Narratives: The Far Right and Pseudo/Anti-Science**

Across the world, we see how people are turning (or are turned) towards nationalist, xenophobic, ultra-conservative, and/or authoritarian movements, parties, and leaders. In their more authoritarian versions, such movements may rely on more or less repressive measures to reign in or crack down on dissent and critical voices, but ultimately even these movements or leaders need some kind of societal support structure. Hence, in their efforts to capture and harness emotional support, many of these movements (and leaders) channel and govern emotions in their broadest sense to reach an audience increasingly beset by securing its everyday existence. Behind this turn towards the far right seems to be a belief that such movements can somehow solve political, economic, cultural, and ideological uncertainties by providing simplified solutions to complex questions. In their nationalist version, they do this by providing a narrative (often in terms of a fantasy snapshot) of the state (and the nation) as stable, uniform, and strong in order to encompass anxiety, neutralize anger, and relieve guilt, while also fulfilling imagined needs for pride, attachment, and desire (Steele and Homolar 2019, p. 214; Browning 2016, 2019; Kinnvall 2018; Mälksoo 2016; Subotić 2016). Insecurity may be one of the most general conditions of human life, and one that is always intimately tied to inequality, social justice, and violence. For some people, especially those whose lives are marked by wars, displacement, urban marginalization, and the effects of climate change and, more recently, the COVID-19 pandemic, insecurity is an ever-present potentiality and experience. However, insecurity is not only about structural (economic, political, and social), epidemic, and environmental realities, it is equally about the narratives, images, and fantasies conveyed through media and political rhetoric about these real or perceived realities.

This is where the notion of ontological security comes in, a concept introduced by psychoanalyst R.D. Laing (1960) and developed by sociologist Anthony Giddens (1991) to account for the effects of late modernity on people's sense of security. Giddens refers to ontological security as a 'security of being', of creating a feeling of a whole and autonomous self. However, in reality, it is a process of 'becoming', as the strive for ontological security is always only a temporary and incomplete process of closing down particular narrative imaginations, fantasies, and desires in order to feel secure in the here and now (Browning 2019; Kinnvall and Svensson 2022). In this, reality and fantasy are always co-constituted, as 'fantasy' is a critical component of world-enactment and conceptualizations of political reality (Sass 2015). From a Lacanian perspective, "fantasy is the narrative frame that constitutes and stabilizes the subjective sense of reality [ . . . ]. Therefore, fantasy captures the process whereby subjects (social actors) relate to and reproduce reality (social structures) by outlining the relationship between subjectivity, social order, and desire" (Eberle 2019, p. 245). In the nativist version of far-right nationalism, fantasies are intimately tied to what Giddens has referred to as a 'sense of place', in which spaces and narratives about certain locales offer important imaginary anchors for political leaders to pin down unknown anxieties amongst the electorate (Ejdus 2017; Subotic 2018; Della Sala 2018). As Kinnvall and Svensson (2022, p. 532) have argued: "Fantasies of past traumas and glories often become 'real' in the hands of far-right leaders, who convey narratives and images of humiliation and shame as well as of pride and superiority to their followers [ . . . ]—thus purporting to ascertain and satisfy a perpetual desire through fantasmatic (fictional) closure and wholeness" (see also Homolar and Löfflmann 2021).

In such versions, fantasies are inadvertently tied up with a masculinist logic and gendered nationalism. Gendered nationalism is thus embroiled in masculinist claims of 'protection', 'manhood', 'imperial loss', and 'mythical pasts', often fueled by the idea of a strong nation that has been weakened by femininization (Nicholas and Agius 2018). "In the imaginaries of far-right populist and center-right movements, this rests on a political ideology that has as its core myth the homogenous nation—a romantic and gendered version of the homeland and homeland culture, both of which act as emotional resources in the appeal to ontological security" (Agius et al. 2020, p. 440; see also Kisić Merino et al. 2021). In the case of the far right, these fantasies often contain narratives of a secure, constant, and reminiscent past as contrasted to an anxiety-inducing present, a present that is often besieged by a fantasmatic projection of the 'other(s)', and "where women are singled out as the symbolic repository of group identity" (Kandiyoti 1991, p. 434). This narrative relies on fantasies of a 'natural' relationship between women and the nation (motherland, home, motherhood), with gender as a 'natural', essentialist dichotomous order (Saresma 2018), in which women act as figurative mothers of the nation-state (Mudde 2019). This relationship between gendered fantasy narratives, ontological insecurity, and the far right is particularly striking in the Indian case of Hindu nationalism and is intimately connected to narratives of pseudoscience: 'fake news', 'post-truths', and 'anti-science' claims.

Populist far-right politicians (from Trump to Bolsonaro to Modi and others)[2] have for years used relativist arguments to discredit overwhelming scientific evidence for anthropogenic climate change, and the COVID-19 crisis has been accompanied with a veritable 'misinfo-demic' (WHO 2020). Hence, a number of extremist politicians have relied on 'truth-subversion' practices to discredit liberal elites, immigrants, and often women. "Anti-science is the rejection of mainstream scientific views and methods or their replacement with unproven or deliberately misleading theories, often for nefarious and political gains. It targets prominent scientists and attempts to discredit them" (Hotez 2021). This suggests that information can be presented in two ways and that each has equal value (CNN vs. the alt-right blog Breitbart, for instance), and relies on a belief that journalists, experts, and politicians are merely representing alternative views on the political spectrum (Kisić Merino and Kinnvall 2022). This signifies a form of emotional blame-shifting in which anxiety is turned into fear and aggression and where gendered fantasies stand out in terms of their nativist origins. It is particularly evident in the Indian case, where the current Prime Minister Narendra Modi has labeled all critique against the government's handling of the COVID-19 crisis as anti-Bharat (anti-India), thus arguing that it is largely an "anti-Bharat conspiracy to create an atmosphere of negativity and distrust in the government" (Kaul 2021). Modi is portrayed here as ascetic, paternal, and efficient—a strongman protector of the Hindu nation—a theme that has deep resonance among the Hindu right, as discussed in the next section.

**Gendered Fantasy Narratives: Hindu Nationalism in Varanasi and the Film Water**

Before Narendra Modi came to power in 2014, Hindu nationalists were waging a battle against resistance movements in Varanasi in their attempts to control the narratives of women's bodies in Hindu religious discourse. In this context, Hindu nationalists' violence against the shooting of the film Water in Varanasi presents an interesting case study by providing the ideological background of Hindu nationalist ontological security-seeking practices and their resistance, while also bringing to the forefront the fantasy narratives of 'motherhood' at the center of Hindutva imaginaries. Located in North India, Varanasi is one of the most sacred places of Hindus, and it is also a highly revered seat of Brahmanical Hinduism. During the 15th and 17th centuries, and under the patronage of landlords, traders, and priest classes in Varanasi, Hindu high culture reinforced the superior position of Brahmins and other hierarchical relationships, including the subservience of women, through the public performance of Ramleela[3], Katha[4], and public recitations of Ramcharitmansa[5] (cited in Freitag 1989, p. 26). Due to these Brahmanical religious rituals and practices, the public sphere in Varanasi continues to be heavily dominated by the

discourse of 'Hindu orthodoxy' and Hindu Brahminic culture, leaving a narrow space for free, secular, and egalitarian thinking[6].

The film Water is based on the deplorable situation of Indian widows in the 1930s. It explores the exploitations of child widows by Brahmin priests in the 'widow shelters' in Varanasi and, upon its release, was an open challenge to the Hindu orthodoxy. The film was believed to hurt the religious sentiments of the Hindu community by depicting Hindu culture in a poor light and by demeaning the cultural heritage of Varanasi and India (*Outlook* 2022; Jakob 2006). Though the film script had been approved by the national censor board, protests erupted on 30 January 2000 by Hindu nationalist groups, such as Rastriya Swayamsevak Sangh (RSS), Vishwa Hindu Parishad (VHP), and the Kashi Sanskrit Raksha Sangharsh Samiti (KSRSS)—a social group directed by the RSS that was formed overnight which specifically targeted the director of the movie, Deepa Mehta (Phillips 2000; Khorana 2009). The protests soon turned violent, and when the filming started, the film crew was attacked, and the sets were destroyed. Here, it is important to note how Hindu nationalist ideology conceptualizes women as protectors of Hindu values and traditions rather than as individuals[7]. In the context of Water, it was also made clear that no matter what the suffering of Hindu widows, this topic should not be opened to public debate (Berglund 2011, p. 90). Depictions of child widows and prostitution challenged and offended the orthodoxy as these 'social evils' were presented as once being part of Hindu customs. Maligning the image of regressive social customs was thus seen as an attack on Hindu traditions by Hindutva proponents.

In this, Hindu sensitivity rests on the Brahmanical patriarchal system, which requires domestication (taming) of Hindu women for the system to be run smoothly—a purpose served by an inegalitarian Hindu nationalist ideology. Thus, for a Hindutva follower, a traditional Hindu woman is a fantasized conformist being (except sporadic examples of women scholars in ancient India), who must show unquestioned loyalties to her husband, family, and the Hindu nation. In the Hindu nationalist imagination, the landmass of India is framed as 'Bharat Mata', Mother India—a patriotic representation of the land in a divine female form. This depiction of India as a Hindu mother goddess has made it a religious duty for all Hindus to worship and protect the nation (Tharoor 2020; Kinnvall 2006; Anand 2011) and is also manifest in terms of how Indian womanhood is revered in the images of many Hindu female goddesses (Kali, Durga, Laxmi). As stated by the BJP spokesperson, Ashok Pandey:

> India is a land of Sita and Savitri . . . women are worshipped as a form of goddess . . . if you want to worship money, worship Lakshmi, if you desire knowledge, then worship Saraswati, Cow is considered our mother, Ganga is also considered our mother. (Tharoor 2020, p. 265)

In the Hindu nationalist fantasy, 'motherhood' is thus at the core of its imaginings—both as nation and goddess(es)—where the cow represents Mother Earth, as it is a source of goodness and its milk nourishes all creatures, while the river Ganges is a personification of the goddess Ganga. Krishna, a central Hindu deity, is often portrayed in stories recounting his life as a cowherd and referring to him as the child who protects cows (*Patheos* 2022). Hence, the Hindu nation/goddess remained untainted before the Hindu nation was conquered, violated, and raped by Muslim and Christian forces. This tendency to imagine the nation as 'pure' is always a gendered strategy related to men's control over women's bodies as women are considered to be defiled or tarnished by other men (Kinnvall 2006, p. 173; Kandiyoti 1991). Notions of community honor thus become contingent upon safeguarding women's sexual purity and domestic roles. As Jitendra Swami, an RSS-affiliated Hindu spiritual guru, asserts: "[T]here is a set model of a character in this country, as a daughter, sister, sisters-in-law . . . we cannot give sexual freedom to women" (quoted in Basu 1993, p. 30). Hindu majoritarianism thus propagates a 'macho-culture' that disregards women's individual agency and their right to make independent choices.

Here, a number of scholars have noted how women are ideally suited to the nationalist project (see, e.g., Sen 2014; Basu 1993), and how "nationalism singles women out as the

symbolic repository of kinship, motherland, or home" (Kandiyoti 1991). Hindu nationalism is thus similar to other nationalisms in its tendency to mythologize tradition and put the burden of old customs on women's shoulders. One important aspect of this longing to return to tradition concerns the resurrection of older forms of family organization and women's roles within them. In this, Hindu nationalism wants to bring back those times when women's sacredness was given importance, a theme associated with Hindu female purity where the female body becomes a site for claiming community homogeneity and order where women "are being re-inforced as essential subjects of the imagined *Hindu Rashtra*" (Bakshi 2020). This control of female sexuality, bodies, and reproduction is crucial to nationalism (Butler and Spivak 2010), in which women become the 'burden of representation' (Yuval-Davis 1997).

For Hindu nationalists, Hindu women thus have an active obligation to protect the nation, and Hindu motherhood plays a vital role in the creation of a Hindu nation. In fact, the role of women for sustaining Hindu nationalism is well-noted (Sen 2014; Banerjee 2012). The film Water hence challenged the Hindutva portrayal of female agency as pious. By portraying the undignified life of widowhood, the film raised critical questions on women's subservient role in the Hindu Brahmanical culture, thereby challenging the discourse of Hindu nationalism and its attempts to control women's bodies. The shooting of the film thus made Hindu nationalists 'ontologically insecure', generating an 'ontological crisis' by questioning the very premise on which 'motherhood' of Hindu nationalism stands. Therefore, the violent protests against the film's shooting can be considered a Hindutva crisis-response to maintain its control over women's bodies through ontological security-seeking practices that eliminate contesting narratives of Indian nationhood. To this day, the narrative of motherhood remains the primary site of female agency and political action in Hindutva ideology (Sen 2014). This became clear in the interviews conducted with individuals and groups resisting the Hindutva narrative. Not only did the interviewees emphasize the linkage between Hindutva fantasies of motherhood and that of patriarchy and the denial of women's rights, but they also clearly related this to violence and fears of critical thinking as a strategy to shutdown Hindutva identity. These are also themes that emerged during the pandemic, as explored later, where narratives of motherhood became connected to Vedic science as pseudoscience and the closing down of Hindutva in response to protest and critical thought.

**Water: Fantasmatical Narratives of Hindutva and Their Resistance**

A number of interviews were conducted with academics, artists, various NGO workers, and feminists who all resisted the hegemonic Hindu Brahminic culture in Varanasi, coming from all castes and religions. During the analysis of these resistance groups, three main themes emerged that illustrate the fantasmatical narratives of Hindutva and how resistance to Hindutva narratives took shape.

*Theme 1: Water Challenged Hindutva Narratives of Patriarchy*

Most of the respondents believed that the 'film set' was vandalized because it challenged Hindu patriarchy. One Hindu women activist believed the film set was attacked because it "*questioned the authority of Hindu patriarchy*", in which "*religion, patriarchy, and Brahminism support each other; if one falls, another will fall . . . people will start questioning their [Hindutva] hegemony . . . if women got empowered, their religion would be demolished*". Similarly, a Muslim female educationist and activist who actively coordinated the resistance movement perceived the vandalism against the film shooting as "*Hindutva attempt to silence progressive and feminist voices*", using "*violence and intimidation*", arguing that "*RSS and Shiv sena demolished the movie set because they find this film against their culture because [the] film shows poor condition of Hindu widows*". In this, she was asserting that all religions tend to oppose change when it comes to women's social reformation and that the RSS is patriarchal, enforcing the idea that women should be confined to their homes. "*[T]hey [RSS/Hindutva] want to impose strict dress-code on women; they have problems with women raising their voices*

[against their oppression], they support widow burning and are in favor of keeping widows in poor situations".

Many respondents also considered Hindutva ideology a serious threat to gender equality and secularism due to its patriarchal nature, where a woman's duty is to produce babies and become a good Hindu nationalist. "*Hindutva proponents do not want women to be educated, otherwise who will cook for them, if women got empowered their religion would be demolished*". Many of the interviewees also stressed that, within the Hindutva framework, a Hindu woman is denied making personal choices on issues of vital concern to her life: arranged marriages are preferred over love marriages, inter-caste marriages are despised, and Muslim men may be criminalized (known as a love-jihad) if married to a Hindu woman. This notion of inter-caste or cross-religious marriages link patriarchy to fantasmatic narratives of the 'nation as mother' (Bharat Mata), in which only pure Hindus (produced and reared by Hindu mothers) are to be considered true members of the nation.

Another respondent, a Christian priest, activist, and educationist, believed that Water was attacked because it did raise serious women's issues, such as "*forced prostitution of widows, child marriage, and their exploitation by the Hindu priests*", thus arguing that "*religion helps to sustain patriarchy since women follow and support regressive Hindu social customs*". Interestingly, most Hindu rituals promote patriarchy and women's subordination by confining them to a lower position in society. Festivals such as Karva chauth, Tij, and Bhaiya Duje require women to keep fasting for the male members of their family. Many such festivals and rituals are hence crucial for the transmission of religious heritage and cultural norms from generation to generation, thus strengthening a patriarchy sustained by the Brahmanical system and Hindutva proponents (Tewari 2007, p. 42). In contrast, this priest said that Varanasi is not just representing Brahmin culture, but it is also a city of composite cultures (Sanjha sanskriti) where dhobis (washermen), mochis (cobblers), and street vendors live. Here, it is important to note how Ganga-Jamuni tajeeb, or the mixed-culture of Varanasi, is known for its tolerance and inclusiveness (Upadhyay 2010). However, this Christian priest was also targeted by Hindutva proponents and was arrested during the Anti-CAA (Citizen Amendment Act) protests in 2019. Other respondents stressed how the "*film highlighted the nasty lives of Brahmins who claimed to live a pious life ... *" and how the film showed "*a historical fact when child widows were exploited and Dalit women were tormented*". Hence, by creating violence against the film's shooting, RSS and Hindutva forces were trying to impose their narrative hegemony over the society—a form of social control that works as an ontological security-seeking practice in the face of resistance.

*Theme 2: Hindutva Is Fundamentally against Women's Rights: Narratives of Good Motherhood*

This control also extends to other choices of their lives, particularly to widows of all ages: "*they have made the rules for widows: what they shall eat; they were forbidden to use salt and garlic so they will not sexually get aroused, they have to wear white dress, had to shave their head ... their food, lifestyle, dress, everything was controlled*"; furthermore, "*they must not look beautiful; they were treated like slaves*". The portrayal of widows in the film Water is similar to this respondent's narrative: widows in shelter houses had to live by the rules coded in the Manu-smriti, they had to shave their heads, wear white dresses, not eat sweets, and were not allowed to meet their male relatives. This is what Hindutva refers to as 'Hindu culture', where women are not allowed to make even the smallest decisions on their lives. "*Even, their 'thought process' are controlled by instructing them to keep themselves busy in the worship of god*". An ideal woman must be a good mother, daughter, or sister, and she must stay pure as the sacred Ganga. In traditional Hindu society, women were discouraged to study and work and denied property rights. Hindutva ideology, similar to other far-right ideologies, imposes this model of society on Hindu women. In this process of domination of women's bodies, Hindutva, as one respondent claimed, "*employs a selective reading of Smriti* [a body of Hindu text] *to validate women's lower position in the Hindu society*". In contemporary times, it invokes the nostalgia of this mythical past by contrasting a Hindu women's character with the Goddess Sita, the obedient wife Savitri and Sati. As Jitendra Swami, an RSS-affiliated

Hindu spiritual guru, lamented: "*even battles in Ramayana and Mahabharata were fought to save the dignity of the women, showing how much women were dignified*".

Such narratives are in line with the Hindutva ideology towards women. Violence against Water was thus an attempt by Hindutva forces to silence those challenging an unfair and unjust oppressive system. Most of the Hindutva proponents are also very critical of the human rights discourse, which is viewed as empowering women and making them challenge social dogmas and evil customs. In the words of one activist, "*Hindutva forces are afraid of [human rights] activism; they are afraid of critical thinking, they are afraid of being challenged*". Hindi media in Varanasi also provided coverage to Hindutva forces and ignored the resistance's protests, while the BJP state government was afraid to offend its core Hindu voters by allowing the film to be shot. Additionally, local Hindutva leaders saw this (opposing an anti-Hindu film) as an opportunity to polarize people—on the lines of Hindu versus foreign culture—to gain political benefits. In addition, the resistance movement—which consisted of city-educated elites—failed to connect and convince the semi-urban people of Varanasi that this film was for their own good. Hindutva forces were thus able to use imaginaries of a mythical past to stir feelings of ontological insecurity among the Hindu religious community. It did so by turning to fantasies of 'motherhood' to discard multiple 'forces of resistance'.

*Theme 3: Resistance Is Possible*

Protest in support of Water has galvanized anti-Hindutva forces who saw the attack on the film set and subsequent violent outbursts by the Hindu orthodoxy as an attack on the composite culture of Varanasi. Feminist NGOs and academic intellectuals co-related this incidence with the violation of women's rights and the curtailment of freedom of expression. They came out in support of Deepa Mehta, the director of the film, but found themselves fighting against Hindutva forces and the ruling Hindu government of the BJP. Actual resistance against Hindutva groups started when the film set was damaged; soon, a public forum, Sanjha Sanskriit Manch (Composite cultural forum), was formed, which tried to problematize the dominant culture of Varanasi, saying that it "*is a composite (syncritic) culture—not Brahmanical culture . . . culture is not monolithic, culture is not fossilized*". One protestor referred to the resistance as a 'civil society movement', supported by writers, artists, and academics " . . . *we were about 200–300 people, this was a small civil society movement with limited resources . . . RSS used religion to mobilize people, used religious symbols and aggressive nationalism that have made people support them*". Soon, resistance was unified, organized, and coordinated: "*whenever there was an attack by Hindutva forces, we went there to protest against them, we have organized protest and sit-ins against such forces*", asserting that, "*we want to tell the people of Varanasi that this is an attack on our composite culture and we shall raise our voices against such forces . . . they* [RSS] *are people who are doing mob lynching and love jihad*".

Despite personal attacks, threats of arrest, and discouragement by the local government and the police, resistance groups continued to participate in protest marches, sit-ins, and submitting petitions to the government. Local media, the police, and the administration stood by Hindutva groups, though, pressurized by the BJP state government: "*the government did not help in creating an environment for the film shooting . . . local media sensationalized their protest*". None of the political parties came to their support, not even those who were secular and socialist. The resistance movement also had limited resources and did not receive much support from the masses. Lacking the emotional discourse of Hindutva, which was able to mobilize on religious lines in a deeply religious society—using emotional narratives of motherhood—the protestors failed to connect with the masses.

RSS and Hindutva groups also have well-established networks to start riots (Brass 2005), which they employ to push their political agenda from time to time. However, as one protester claimed, "*against all odds, they fought well*", and the resistance resulted in feminist Delhi-based NGOs lending their support to the resistance movement. In addition, theater and arts organizations made people aware of the violence perpetrated by Hindutva

groups against the film crew. Hence, resistance in support of Water represented a struggle against oppressive fantasmatical Hindu customs which discriminate and dehumanize Hindu widows. However, resistance was also clubbed together with other issues of equal concern, such as safeguarding the composite culture of the city, freedom of expression, and secularism. Those who fought believed in religious pluralism, human rights, and gender equality. Through resistance, progressive and rebellious voices shared the cultural space of the city. Resistance thus meant different things for different struggling groups. For feminists, it was a matter of women's rights, for some it concerned freedom of expression and secularism, while for others, it was a struggle against 'untouchability' and 'manu-smriti' (the laws of Manu).

Although the resistance movement was not able to achieve its goal (as the shooting of the film was prohibited), its symbolic achievement was remarkable, and it led to a proactive discussion on less talked about sensitive issues, such as the exploitation of Hindu widows, and drew critical attention nationally and globally. Support for Water shows the possibility of resistance against hegemonic Hindutva, even in the most sacred of cities. Interestingly, in other parts of India, such as in Sabarimala, Kerala, women successfully gained entry into a Hindu temple by defeating Hindu orthodox forces (Karindalam 2019). In Maharashtra, the Shani Shingnapur temple lifted the ban on women's entry due to feminists' struggle for gender equality (*The Hindu* 2016). Such examples show how resistance against fantasmatical hegemonic discourses can be won by questioning ontological security-seeking practices that aim to shut down Hindutva.

Since the Water controversy, resistance against Hindu nationalism has continued in various parts of India. Before COVID-19 struck India in March 2020, the award wapsi[8] movement against Hindutva intolerance occurred in 2015, the Bhima Koregaon/Elgar Parishad[9] case/movement in support of Dalits' dignity took place in 2018, and the 2019 anti-CAA (Citizen Amendment Act) protests posed tough resistance against the Modi government's policy to demonize Muslims and to suppress Dalit resistance movements. During the COVID-19 pandemic, farmers' protest to repeal the 'three bills' became a nightmare for the Hindu nationalist government. Along with it, voices of resistance emerged from intellectuals, journalists, ex-bureaucrats, and students against the Modi government in relation to mismanagement of the COVID-19 pandemic. The government's response to much of this resistance has been to suppress and discredit it as 'anti-national', while promoting daily pseudoscientific fantasy narratives of Hindu nationalism for dealing with the coronavirus. Of particular importance is the extent to which such pseudoscientific fantasies were based in narratives of motherhood and how they, during the pandemic, worked as ontological security-seeking practices in their connections to mythical pasts and ideological underpinnings, as outlined in the Water section. This is dealt with in the next two sections, which start with an overview of the resistance against the Modi government and then proceed to a discussion of Hindu nationalists' (in line with and beyond the Modi government) fantasmatic responses to the pandemic. These two sections are mainly based on newspaper materials and speeches, although some of the interviews are used to highlight the link between fantasmatic narratives of motherhood and Vedic science as pseudoscience.

**Resistance during the Pandemic**

The world's largest demonstration and probably the biggest protests in human history (as reported by *Time magazine*, Bhowmick 2021) started against the Modi government in August 2020. As a protest movement, it seriously challenged the Hindutva agenda of benefiting big corporates (such as Ambani and Adani corporate houses) at the expense of poor Indian farmers. The 2020–2021 farmers' protest, which triggered countrywide demonstrations, was against three farm acts passed during the pandemic by the Modi government in September 2020, without any serious debate or consultation. The government claimed that these laws would benefit the farmers, while farmers argued that they may lose their land to big corporations and encounter the loss of government subsidies. The farmers' protest

started in Punjab and Haryana but soon spread to other parts of India (Delhi, western Uttar Pradesh, Bihar, Hyderabad) and attracted support and participation from all walks of people, from intellectuals to students, to ordinary people and national politicians. The protests not only galvanized support from farmers within India, but also drew international support, including American pop star Rihanna and Swedish environmental activist Greta Thunberg (Arvin 2021). Farmers, in some parts of Haryana and Punjab, even warned BJP leaders not to enter their villages, and protesting farmers jammed national highways and went to West Bengal to campaign against BJP in the state election. The victory of Mamata Banerjee against BJP in West Bengal represented a political resistance against a growing and hegemonic Hindu nationalism. The farmers' protest thus emerged as one of the largest symbols of resistance against the Hindu nationalist party.

Apart from large and organized resistance movements, there has been sporadic resistance by individuals and professionals in response to Hindu nationalist policies during the pandemic. The nature of such protests has been rejecting, denying, or questioning Modi's polices during the pandemic. Those who criticized the government's policies were penalized in various ways, however, from detention to framing sedition charges. Dissidents were charged under India's Epidemics Diseases Act, and the 2005 Disaster Management Act. Both provide for possible prison terms and fines. Numerous arrests of activists and scholars were carried out under the National Security Act (NSA), the Public Safety Act (PSA), and the Unlawful Activities (Prevention) Act (UAPA) (Alam 2020). Complaints were filed against several journalists seen as being critical of the state's pandemic response, especially the manner in which lockdown was imposed.

Later, as a second wave of the pandemic engulfed India in April–May 2021, the number of COVID-19 deaths drastically increased and the demand for medical supplies and vaccines shot up. People and activists started posting desperate messages on social media asking for oxygen cylinders, hospital beds, and ambulances. Consequently, on twitter, hashtags such as #Modiresign, #ResignModi, and #DisasterModi went viral (Ittefaq et al. 2022). An online petition on Change.org collected more than a million signatures demanding Modi's resignation. Suddenly, public sentiments started to turn against the Modi government. In addition, constant critical coverage by international media set the public narrative against the Indian government, which consequently resulted in the lowest rating of Narendra Modi since he came into power in 2014 (Mukherji 2021). Rather than addressing the grievances of its people, the Modi government saw this as an attempt to malign its image. In response to this 'public narrative' of gloom and doom, the Modi government tried to silence dissent through various means.

Under the government direction, fifty-two critical comments from twitter were removed (*Asia News* 2021). Cases were filed against those asking for help on social media. The Uttar Pradesh Chief Minister, Yogi Adityanath, directed officials to take action under the National Security Act and seize the property of individuals who spread 'rumors' on social media, claiming that hospitals were struggling to maintain their oxygen supplies. In Delhi, however, due to the shortage of vaccines and the critique of Modi's vaccination strategy, the Aam Aadmi party (AAP) put up satirical posters, asking, "*Modiji, why did you send vaccines of our children to foreign countries?*". The Delhi police responded by arresting those who had put up the posters, most of them being auto rickshaw drivers, daily wage laborers, and unemployed people. The AAP party came to their support, and a wave of resistance against the government started on social media (Bhardwaj 2021). Suddenly, many people, including political workers and the Congress party leader, Rahul Gandhi, shared photo shots of concerned posters on their twitter handles, challenging the government to arrest them.

Interestingly, Indian courts also became part of the resistance against the Modi government, extending their support to dissenting voices. The Supreme Court instructed the government not to punish anyone criticizing the government on social media for the situation concerning the pandemic. The Madras High Court questioned the central government for not doing enough to prevent the pandemic and the Delhi High Court issued a notice of

contempt to the government for defying its order to supply adequate oxygen to more than forty New Delhi hospitals (Krishnan 2021). The Allahabad High Court in BJP-ruled Uttar Pradesh, while holding the state accountable, equated the death of COVID-19 patients due to a lack of sufficient oxygen supplies in the hospitals as a 'criminal act and no less than a genocide' (*Asia News* 2021). In response to these criticisms (especially the lack of medical supplies—vaccines, hospital beds, oxygen cylinders), the Hindu nationalist government invoked a mythical imagination of Hindu lifestyles during ancient India when people were healthy and strong. A fantasy narrative was created, within which Hindu populist imaginings, such as the infallibility of indigenous medicines like cow dung and cow urine, were contrasted with modern Western medicine. Based on Hindu belief systems of 'Vedic science', these myths were pushed as a cure to modern diseases, such as the coronavirus. The next section discusses how, through the promotion of a particular kind of pseudo-science that takes 'motherhood' as its defining departure, Hindu populist imagination was being re-memorized and glorified in opposition to modern scientific knowledge.

### Fantasy Narratives of Motherhood and Hindutva Pseudoscience

On 3 April 2020, the Prime Minister Narendra Modi urged Indian citizens to light a candle/torch for 9 min at 9 pm. Minutes later, social media flooded with posts about how this move would eliminate the coronavirus. The BJP leader S.A. Ramdas claimed that lighting candles as instructed by Prime Minister Narendra Modi would kill the coronavirus as the virus would die due to the heat produced by the candles (Bharati 2021a). Modi's intellectual supporters also made claims that traditional 'Shankkanaad' (blowing of the conch shell) could kill the virus (Mohapatra 2020). Modi and his supporters thus reproduced a particular kind of far-right populism—through Hindutva pseudoscience—that challenged the supremacy of modern science and affirmed the hegemony of Hindu cultural fantasies of a gendered 'glorious Indian past'. Hence, Modi's appeal to the public to stay at home for twenty-one days at the beginning of the pandemic made references to Bharat Mata, by connecting the motherhood of the nation to the fetus that stays in the mother's womb for nine months (Radhakrishnan 2020). *"One step out of your door, beyond the 'lakshmana rekha', can bring in this deadly disease to your home"* (Modi, in his address to the nation during lockdown, cited in *ibid*), where the 'lakshmana rekha' refers to events in the Ramayana in which the goddess Sita is not to venture beyond the lines drawn by Lakshmana, and when she does, is kidnapped by Ravana. In this blurring of modern and Vedic science, Modi himself has often remained the stern leader, leaving his associates to interpret his actions as evidence of Vedic superiority over Western knowledge and medicine (see Subramaniam 2021). Here, Sen (2020) argues that the fear of the disease has consistently resulted in the invocation of the mother goddess, described as 'Corona Mata'—the goddess of pestilence—who has the power to tame the disease. This goddess is not one, but several deities invoked as 'Amman' (a popular goddess in Tamil Nadu) and is there to heal the disease and provide ontological security at times of distress.

The Ministry of Information and Broadcasting, funded by the BJP government, was also quick to re-telecast two mythological series from the 1980s during the lockdown, Ramayan and Mahabharat, whose epical contents form the basis of Hindutva politics. The series include characters in ornate costumes, fantastical creatures, and depictions of different gods, and emphasize female characters as being subservient to male ones and the glorification of masculinity (Devani 2020). In line with the analysis of Water, the series can be seen as a prime example of how, through a daily fix of entertainment, the government were to instill Hindutva values of Bharat Mata (Mother India) as the basis for pseudoscientific beliefs.

Narratives of pseudoscience rooted in Hindu nationalism have been on the rise since Prime Minister Narendra Modi came to power in 2014. Modi's affirmation in 2014 that the transplantation of the elephant head of the god Ganesha to a human body was a great achievement of Indian surgery, reflects an attempt to recover the 'lost glory' of the Vishwa guru Bharat (Rahman 2014). It seeks to temper rational modern science with

Hindutva nostalgia of the superiority of the Vedic sciences, in which science and technology mediate mythological and divine worlds (Subramaniam 2021). Modi's mockery of science also paved the way for his supporters to narrativize a fantasy of Hindu pseudoscience in establishing the hegemony of Hindutva and polarize people on the lines of rational and religious modes of reasoning. Employment of pseudoscience thus acted as a way to deal with the 'ontological crisis' of Hindu nationalism during the turbulent times of the pandemic. Hindu nationalists invoked 'ontological security' by appealing to the nostalgic greatness of ancient Indian science to cure modern diseases, where the cow as mother (India) became an important ontological security-seeking practice for Hindutva nationalist fantasies.

Such mythical beliefs of Hindutva pseudoscience led the BJP science minister Harsh Vardhan in 2017 to fund research to validate the idea that *panchagavya*, a concoction that includes cow urine and dung, is a remedy for a wide array of ailments. During the pandemic, such Hinduized narratives of 'pseudoscience' took on a hegemonic form when BJP ministers—one after another—started to assert the benefits of cow dung and cow urine in curing and preventing the coronavirus. The BJP member of parliament Pragya Thakur asserted, for instance, that 'Gau-mutra ark' (cow urine extract) of a desi cow could prevent lung infection, claiming that she would use the 'gaumutra ark' every day and did not take any other medicine for the coronavirus (*The Free Press Journal* 2021). In another such incidence, the BJP Minister Usha Thakur claimed that performing 'Yagna Chikitsa' (treatment through Hindu offering rituals) could ward off a third wave of the virus and that adopting the Vedic lifestyle could serve as a protection from it (Bharati 2021b). Although doctors have dismissed such claims (Dave 2021), not only BJP ministers, but also their supporters and influential personalities discarded the efficacy of modern medical sciences over ancient Hindu medicinal knowledge.

The spiritual guru Ramdev blamed, for instance, allopathic medicine for coronavirus-related deaths, claiming that 'Coronil'—a product of his herbal pharmacy—could cure COVID-19 infections in seven days (Ray 2021). Interestingly, the Indian government, through its Ayush ministry[10] which promotes alternative medicines such as Ayurveda, granted a license to Coronil. However, doctors of the Indian Medical Association have challenged such claims and have taken legal action against Ramdev (Ani 2021). Ramdev also claimed that putting mustard oil through the nose would push the virus into the stomach where it would die due to acid (Roy 2020). In the BJP-ruled state of Gujarat, a forty-bed state hospital in a Goshala (cow shed) was treating COVID-19 patients with cow dung and urine while chanting mantras (Desai 2021). Patients were also covered by a layer of cow dung as part of the treatment. The sacredness of the cow—not least the cow as mother—has thus been reasserted as a symbol of growing Hindu nationalism. The idea that the cow's milk, dung, and its urine have healing and anti-septic properties and can cure any disease takes its inspiration from religious beliefs in Hindu cultural superiority over modern science. During the pandemic, a Hindutva cultural narrative was hence set in motion through the employment of the efficacy of Ayurveda, cow dung, urine, and other Hindu religious cultural properties and practices.

These Hindu cultural narratives were constructed to deal with Hindutva ontological insecurity to counteract growing dissatisfactions under the BJP government concerning its handling of the pandemic. In this context, when hundreds of people were dying on a daily basis due to the lack of oxygen, hospital beds, medicine, and vaccines, and with the Modi government failing to cater to their desperate needs, Hindutva was desperately seeking an ontological security—a fantasy cover—in pseudoscience in order to provide an immediate psychological healing to its Hindu citizenry. Here, it is important to note how Hindu nationalists not only believe in the supremacy of their race, but also in their cultural knowledge, such as Ayurveda and Vedic science. Ayurveda has been recognized as an alternative medicine but has not been accepted as a mainstream medical system by the Indian Medical Association or by the Western medical world. Hindu nationalists frequently cite herbal antidotes for modern diseases based on a nostalgia rooted in the superiority of

the Vedic knowledge (Subramaniam 2021; Radhakrishnan 2020). Though pseudoscience has been a pet-project of Hindu nationalists, during the COVID-19 pandemic, it became a Hindutva tool to counteract bad publicity of the Modi government in the Western media. However, it can also be seen as a reaction to Hindutva's own ontological insecurity. The Modi government underestimated the disaster a pandemic could cause to the people, making Hindu nationalists' emotional call to employ Ayurveda or herbal medicines an attempt to hegemonize scientific narratives under the narratives of Hindu pseudoscience, in which mother goddess and Brahat Mata became one.

Interestingly, the popularity of such pseudoscience seems to be on the rise among Indian citizens, as recorded by google data[11]. Google searches for 'Coronil' went up significantly amid an unprecedented surge in COVID-19 cases, showing how people were indeed influenced by the discourse of Hindutva pseudoscience. Hence, Hindu nationalists were able to nativize an 'ontological security crisis' (due to the COVID-19 pandemic) onto the promotion of Hindutva pseudoscience, which falsely provided some people with a sense of temporal ontological security. These Hindutva fantasy narratives are anchored in beliefs rather than logics and facts. By boasting of the antiquity of Indian medicine and its superiority, Hindu nationalist have sought to invoke a nostalgia about the glorious past when Bharat was vishava guru, and still can be. Hindutva 'pseudoscience' is thus yet another 'fantasy narrative' meant to establish its intellectual superiority and cultural hegemony. Nevertheless, resistance against Hindutva 'pseudoscience' has continued. Indian doctors have constantly warned people not to use cow dung to cure COVID-19 (Pleasance 2021). "There is no concrete scientific evidence that cow dung or urine work to boost immunity against COVID-19, it is based entirely on belief", said Dr. J.A. Jayal, president of the Indian Medical Association (*ibid*.). Senior scientists have also resigned over ignoring scientific data in dealing with the pandemic. Expressing their anguish, Indian scientist Swarup Sarkar lamented that "evidence and science were selectively neglected" (Financial Times 2021).

Alternative media[12] have persistently challenged the Hindutva fantasy narrative of pseudoscience, whereas citizens and intellectuals have resorted to Twitter and Facebook to criticize Hindu nationalists' attempts to impose such fantasies over the nation. Government responses to this resistance have been quick. A recent example can be found in the BJP government in the state of Manipur, which slapped the National Security Act (NSA) on journalists for Facebook posts that point out that cow dung or urine cannot cure COVID-19 (*The Indian Express* 2021). Many prominent intellectuals, fighting against superstition and pseudoscientific ideas and practices, have also been murdered by Hindu fundamentalists, including Narendra Dabholkar, a physician, and M.M. Kalburgi, a former vice-chancellor of Kannada University in Hampi (*ibid*.). Overall, the promotion of pseudoscience can be seen as another 'fantasy narrative' in the arsenal of Hindu nationalism to deal with the current 'ontological security crisis'. Invoking the therapeutic benefits of cow dung and cow urine to cure COVID-19 suggests that Hindu nationalists are seeking an ontological security within the fantastical imaginings of its indigenous traditions of motherhood. However, voices of resistance against it are also growing stronger, as the case of the film Water suggests, together with numerous protest movements during the last few years.

**Conclusions**

It is clear that Hindu nationalists employ a number of fantasy narratives in order to institutionalize their own hegemony and create a temporary narrative closure of ontological security—both to the movement itself and to the followers. The gendered implications of such fantasy narratives are particularly clear. Not only is India, or Bharat, viewed as a mother goddess, in which pious images of mythological women are held up to hail and revere, but also cows are similarly put on a pedestal of sacredness in their motherly features. Providing milk and nurturing generations of Hindus, the cows are portrayed along motherly lines of the sacred nation. Protests against these Hindu nationalist imaginings have encountered severe pushback from far-right Hindu nationalist groups as well as from

the BJP-led government, not least through cow vigilantism. For Hindu nationalists, Hindu women thus have an active obligation to protect the nation, and Hindu motherhood plays a vital role in the creation of a Hindu nation—a Bharat Mata. By promoting Hindutva pseudoscience in response to the ontological security crisis created by the COVID-19 pandemic, Hindu nationalists have further relied on mythical Vedic knowledge to provide their movement and followers with a sense of temporal security and safety. The government's response to most of this resistance has been to suppress and discredit all resistance as 'anti-national' while promoting daily pseudoscientific Hindutva fantasy narratives of motherhood for dealing with both the film Water and the coronavirus.

**Author Contributions:** Writing—original draft preparation, C.K. and A.S.; writing—review and editing, C.K. and A.S. All authors have read and agreed to the published version of the manuscript.

**Funding:** This research was partly funded by The Swedish Social Sciences Research Council for the project: Divine Ganges, Profane Development: Sacred Geographies and the Governing of Pollution. Project number: 2019-04279.

**Informed Consent Statement:** Informed consent was obtained from all subjects involved in the study.

**Conflicts of Interest:** The authors declare no conflict of interest.

## Notes

[1] Interview subjects were selected by consulting the local library in Varanasi. A documentary analysis was also conducted at the local library at Visesherganj and at a Benares Hindu university library. Crucial information about the film Water and the chain of events and persons involved in the controversy were gathered from the Hindi daily, *Daink Jagran*, a right-wing newspaper. Using the information from newspapers as a sampling frame, interviews were arranged, using a semi-structured interview guide. The interviews also focused on subsequent issues of Hindu nationalist significance, such as cow vigilantism, love Jihad, and numerous communal riots under the Modi regime, as well as on reactions to the COVID-19 pandemic. The researcher also observed socio-political discussions and engaged in informal chats in public spaces, such as tea stalls, public parks, and coffee houses, taking notes of relevant discussions of Hindutva politics, as well as of informal resistance to the current government's policies.

[2] The popularity of Modi differs from many other far-right leaders, however, and the 'Mood of the Nation' survey conducted annually between 2016 and 2021 consistently listed Modi as the most popular prime minister among members of the public (India Today 2021).

[3] Ramleela is the story of Ramayana (Hindu sacred book) performed in public.

[4] Katha is a recitation of Hindu religious scriptures and rituals.

[5] Written by Tulsidas, Ramcharitmanas is considered one of most scared book of Hindus, discussing ideals of society and setting the role models of all aspects of Hindu society.

[6] Patronage of Ramcharitmanas activities from Ramlila to the katha is particularly explicable when seen not only as an auspicious act of charity, but also as an investment in the form of didactic instruction for the lower-caste residents of the city dominated by these power holders. See also (Lutgendorf 1989).

[7] A current example of Hindu orthodoxy is violent protest against the entry of young women in the Sabarimala temple. Going against the verdict of the Supreme Court's ruling in favor of the entry of women in the Sabrimala temple in Kerala (South India), RSS and its affiliates waged a violent protest. In response, a large number of women formed a huge human wall (as a message of gender equality and women's empowerment) to support women's entry in the temple (see Vijayan 2019).

[8] The 'award wapsi' movement in 2015 refers to when more than 50 writers returned their awards to protest alleged growth in intolerance under the Narendra Modi regime.

[9] The 2018 Bhima Koregaon violence refers to violence during an annual celebratory gathering on 1 January 2018 at Bhima Koregaon to mark the 200th anniversary of the Battle of Bhima Koregaon. The violence and stone pelting by the crowd on the gathering resulted in the death of a 28-year-old and injury to five others.

[10] The Ministry of Ayush is a ministry of the Government of India, responsible for developing education, research, and propagation of indigenous and alternative medicine systems in India.

[11] https://trends.google.com/trends/explore?date=today%203-m&geo=IN&q=coronil. Accessed on 5 January 2022.

[12] https://www.youtube.com/watch?v=QqZpp8jlvp0. Accessed on 7 March 2022.

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
