# Peer review of "Enforcing and Resisting Hindutva: Popular Culture, the COVID-19 Crisis and Fantasy Narratives of Motherhood and Pseudoscience in India"

_socsci, doi:10.3390/socsci11120550_

Round 1
Reviewer 1 Report
Review of ”Resisting Hindutva: Popular Culture, the Covid crisis and fantasy-narratives of gendered bodies in India”
Thanks for giving me the opportunity to review this piece. Before beginning my review, I want to make clear that I am not an expert on the empirical context of India and Hindu nationalism, and therefore my comments will be limited to conceptual and methodological issues, as well as questions of structure and presentation.
This article discusses the use of fantasy narratives by Hindu nationalists, and especially how such are used in response to actors/narratives that resist or question the legitimacy of Hindutva ideology or Hindu nationalist rule. The analysis draws on a theoretical framework of how far-right actors deploy gendered fantasy narratives to provide ontological security. The main part of the article consists of two case studies, the first studying controversies around the screening of the film Water in 2000 in the city of Varanasi, and the second focusing on how Hindu nationalists have deployed pseudo-scientific arguments in response to the Covid-19 crisis, and how such arguments have been resisted. In conclusion, the authors argue that the two cases illustrate how Hindu nationalists use gendered fantasy narratives to institutionalize hegemony and provide ontological security.
This article provides a fascinating insight into the role of gendered fantasy narratives by nationalist movements, engaging with two quite different cases which however display similar elements. The theoretical focus on gendered fantasy narratives and ontological security is excellent and the authors convincingly show how these logics are at work in these two cases (although see my third point below!). While discussing issues on a (sometimes) high theoretical level, the presentation is clear and easy to follow. I can clearly see how this piece contributes to the Special Issue as well as to a broader literature on ontological security and the far-right, especially its gendered aspects.
Before recommending publication, I would suggest a few revisions which I believe together would clarify the contribution of this article.
First, I think the author could be clearer and more consistent when spelling out the empirical focus of the article. In the current version, the focus is presented somewhat inconsistently: whereas the title signals that the article focuses on how Hindu nationalism is resisted, the abstract as well as aim (lines 50-52) rather suggest that focus is on how such anti-Hindutva resistance is counteracted by Hindu nationalists. In the analysis, both these issues are addressed (i.e. resistance AND counteracting resistance), which I believe provides a more comprehensive picture. I would therefore suggest that the authors check how the focus is framed in (mostly) title, abstract and introduction so that the reader is given a fair impression of what is to come.
Secondly, and my most important point, I believe the authors could do a better job in explaining the rationale for including these two particular cases and what insights can be gained from studying them, individually and (more importantly) together. The two cases are different in many ways – local controversies around a film screening more than two decades ago vs. recent reactions to government responses to a global pandemic. That is not a problem in itself, but the authors should more clearly articulate what we gain by looking at both cases, that could not be gained from only looking at the second case (which is more clearly related to the topic of the Special Issue). This is mainly a matter of better introducing the two cases and the motivation for studying them in the introduction.
Thirdly, there is room for a more consistent and thorough application of the theoretical framework. In the first case study (the film protests), the focus on how fantasy narratives contribute to ontological security tends to disappear. In the second case study (covid-19) the gendered elements could be brought out more explicitly.
Fourth, some more details could be provided about methods and material. The article draws on two quite different sets of methods, for the first case study interviews with protesters, and for the second case study mainly international media coverage. I suggest a paragraph on methodology is added – and especially to what extent the different methodologies matter for the results.
To conclude: although I believe these suggested revisions are important (again, I especially want to emphasize the second!) I consider them relatively minor and they should not necessitate a major restructuring of the article. I wish the authors good luck and look forward to reading the finalized article.

Author Response
We would like to thank the three anonymous reviewers for their excellent comments which we have addressed in as much detail as possible.
All the reviewers note that the empirical aim of the article is slightly inconsistent as the focus on resistance in the title and in the abstract does not give proper attention to how resistance is counter-acted by Hindu nationalists. They also note that the special issue is on the pandemic and resistance and would like clarification as to what role the film Water and Varanasi play for the general argument of the special issue. These are all very valid points which we have addressed in relation to the reviewers’ comments.
Reviewer 1.
Note. 1. The above is noted by reviewer 1. We have changed the title of the article to ‘Enforcing and Resisting Hindutva’ and have spelt out in detail that we deal with how Hindutva is enforced through narrative fantasies of motherhood as well as how such narratives have been resisted in both the case of the Water and during the pandemic.
Note 2. Reviewer 1 notes that the rationale for addressing the two cases needs to be spelt out. We fully agree. We have now thoroughly revised the article to show how the Water case provides an ideological background of Hindu nationalist fantasies of motherhood and that the two cases are not chosen for comparative reasons but to explore and illustrate how Hindu nationalists have consistently used fantasy narratives of ‘motherhood’ as ontological security seeking practices, and how resistance to such narratives has continued to be met by repression and violence. Narratives of the anti-Water Hindutva movement was about reaffirming nostalgia in terms of the sacredness of Hindu women in ancient India and their imposition in the modern context. The movement was also about asserting the superiority of Hindu culture and ancient Indian mythical knowledge. Both of these have re-emerged in the recent pandemic context through a privileging of a pseudoscience that takes its point of departure in narrative fantasies of Vedic science and motherhood. Hence, the article shows how continuity in fantasy narratives can work as ontological security seeking practices over time that reassert dominance and hegemony in the face of resistance.
Note 3. Reviewer 1 also notes that a more consistent and thorough application of the theoretical framework could benefit the analysis, especially how fantasy narratives and ontological security are related in the Water case and how gender seems to disappear in the pandemic case. The section on Water has been significantly reduced and is discussed in terms of how it clarifies the ideological underpinning to Hindutva and how Hindu nationalist forces were able to use mythical pasts to stir feelings of ontological insecurity among the Hindu religious community, by using fantasies of ‘motherhood’ to discard multiple ‘forces of resistance’. The focus on ‘motherhood’ and its relation to Bharat Mata (mother India) and Vedic science – performed as pseudoscience – is now also the main focus in terms of how Hindutva has worked during the pandemic. Resistance in both cases is described in relation to Hindu nationalist fantasy narratives of motherhood; i.e., how Hindu nationalist forces have used such fantasy narratives to close down Hindu identity and enforce repression and violence in the face of resistance.
Note 4. Reviewer 1 would like to see a discussion of methodology. We have added a section in footnote 1 where we describe how the interviews and other techniques have been used and inserted a paragraph in the text of how the two sections rely on different material, although the interviews also dealt with more general issues of Hindutva and ‘motherhood’ that also played a significant role in the section of the pandemic.
Reviewer 2 Report
The aim of this study is to analyze ‘how Hindu nationalists employ fantasy narratives to counteract resistance, with a particular focus on narratives of ‘motherhood’ and ‘pseudo-science’’ (line 70-71). The main strength of the manuscript is its excellent knowledge of far-right Hindu nationalism and the resistances against this nationalism in India as well as the attempt to discuss the narratives that are used by Hindu nationalists during pre-covid and covid pandemic times.
General comments:
The manuscript is, as far as I understand it, submitted with the intend to get it published in a special issue on ‘Narratives of Resistance in Everyday Lives and the Covid Crisis’. A weakness of the manuscript is that the aim and the data do not explicitly focus on covid and everyday lives. The aim has a focus on narratives by Hindu nationalists, whereas much of the data has a focus on the narratives of their opponents. This becomes confusing as a reader. It means that it is difficult to grasp how the scope of the manuscript fits the call for this issue and how the results contribute knowledge to the research theme.
A third point of improvement is that the reasonably chosen theories (ontological security following Laing and Giddens which is linked to gender, nationalism and the far-right) and method (narrative analysis) are not used in any clear way in the analysis of the data. They are only to certain extend paid lip-service to in the conclusions of sections. Making explicit how the theories and methods are used in the analysis will make the argument and the results more convincing.
The good news, I think, is that there is potential to improve these suggestions.
Specific comments:
- I would recommend rewriting the aim and while doing this considering what the analysis can contribute to the research field. This might mean moving away from a descriptive ‘how’ question that asks you to map developments, to perhaps focusing on more of a puzzle by asking a ‘why’ question or emphasizing a comparative investigation of continuities and changes.
- To clarify the aim, you might consider adding research questions that break down that aim. These could for example address different actors and/or concern narrative struggles about specific themes or the effect of narrative struggles upon narrative power dynamics and social change. This should then be clearly followed up so that it returns you to your aim.
- It would also be good to be more explicit about the data. Who has produced which narrative for which purpose and with what effect? Why are these narratives chosen and not others? How do the different cases of narratives relate to each other? The first two pre-covid cases emphasize gender and nationalism, whereas gender is gone in the third case of the pandemic, despite there being many discussions about for example domestic violence during the pandemic, also in India. Emphasizing the link to the pandemic (pre- or during) might already help, but this also concerns the narratives that are chosen or not. Can you motivate your choices better in relation to your aim?
- Clarifying what is assumed, what specific narratives are being analyzed and what the results are from the analysis might make it easier to emphasize continuities and changes during pre- and during-covid periods. I presume these are important for the theme of the special issue.
- While I was reading, I was wondering whether the film, the city and the pandemic are cases and if you perhaps were using a case study method? A case could contain different narratives and counter-narratives. It made me wonder if this concerns three cases that are being compared or are these three examples of the same case? This can be related to the comment about continuities and changes above.
- When looking over the presentation of the theories and the discussion of the results, it might be good to consider what is generalizable and what not. There are for example important debates about the phenomenon of homo-nationalism in research on gender, sexuality, and nationalism (in addition to the older debates about links between gender and nationalism that you mention) and about the global generalizability of links between gender, the far-right and nationalism in contemporary societies (this generalizability is questioned). What is the contribution of your study to these debates?
- In relation to the method, I was wondering why you did not use a strategic narrative approach or a social narrative approach, since these seem useful for your analysis.
- It would be good if you showed more of your use of a narrative analysis in the presentation of the analysis of specific narratives by specific actors for specific purposes.
- The focus of the aim (as it stands) is on fantasy narratives. Could you please define this? Could you be clearer about the importance of this being fantasy narratives, rather than other types of narratives? This can for example be in the Hindu nationalist strategy to counteract resistance against their politics but also in the narratives of resistance.
- Can you be clearer about what is possible to say about the effects of (fantasy) narratives upon possible power shift dynamics? There is only implicit focus on narrative power struggles and on the possible effects of such power struggles on power shift dynamics, whereas I suggest that this could be emphasized and discussed more explicitly, given your comparison over time.
- As it stands, the link between the materiality of the bodies of widows etc. and the fantasy narratives about male and female bodies is not entirely made clear or shown in the analysis. Could you clarify this? The link between fantasy narratives about mothers, gods and cows are linked to narratives about pseudo-science in the conclusion, which makes this somewhat clearer, but the argument is unconvincing due to the lack of stringency in the analysis. Perhaps some of what is mentioned in the conclusion could be moved to the beginning, after which you could show how you come to this conclusion, by showing more of the analysis?
Author Response
We would like to thank the three anonymous reviewers for their excellent comments which we have addressed in as much detail as possible.
All the reviewers note that the empirical aim of the article is slightly inconsistent as the focus on resistance in the title and in the abstract does not give proper attention to how resistance is counter-acted by Hindu nationalists. They also note that the special issue is on the pandemic and resistance and would like clarification as to what role the film Water and Varanasi play for the general argument of the special issue. These are all very valid points which we have addressed in relation to the reviewers’ comments.
Reviewer 2.
Note 1. Reviewer 2 also notes the inconsistency between the focus of the special issue on resistance and the pandemic and the article’s mixed topic. This has been dealt with through a change in title, focus and a clarification in terms of how the cases are connected. Please see reply to reviewer 1 above.
Note 2. Reviewer 2 also notes that the theories need to be further integrated into the analysis and the methods spelt out. This has been addressed in detail in the revised manuscript. Please see response to Reviewer 1 above.
Note 3 and 4. Aim. Reviewer 2 suggests a change in focus to clarify the aim; addressing why questions instead of how questions and to add research questions to break down that aim. We agree that these are very relevant suggestions, but have chosen instead to clarify the aims and make sure we have a consistent argument that runs through the entire article. This entails a particular focus on the extent to which fantasy narratives work as ontological security seeking practices that reassert dominance and hegemony in the face of resistance in order to close down particular identifications (here Hindutva). In our case the aim is to show how the narrative of ‘motherhood’ has defined Hindu nationalist fantasies over time and how this particular fantasy narrative became linked to Vedic science as ‘pseudoscience’ during the Covid pandemic. In doing this we show how any resistance to such ontological security seeking practices has been met by exclusion, repression and violence, while also inadvertently opening up for resisting forces to redefine this exclusionary narrative to allow for protest.
Note 5, 6, and 7. To be more explicit about the data, continuity/change, and cases. Reviewer 2 points out (in line with Reviewer 1) that gender disappears in the pandemic. The change in narrative focus has addressed this and gender is now at heart of the analysis in both cases and the relationship between the two cases has been spelt out to address the continuity and change dimension. The city, Varanasi, has been toned down and more emphasis is on the ideological underpinnings of Hindutva in the case of Water and how this has re-emerged in the pandemic, as well as how resistance in both cases have been met by ontological security seeking practices that close down exclusionary identifications of Hindutva.
Note 8, 9, 10 and 11. Generalizability, contribution to theoretical debates, why not strategic narratives, use of narrative analysis, and fantasies. These are all very good points. The revised manuscript attempts to clarify how ontological security seeking practices attempt to close down identifications through fantasies – desires of feeling whole, of wanting to be but never quite getting there. This is a contribution to the debate in ontological security studies of becoming over being, but also on how gender and nationalism play into this process. There are numerous other theoretical debates that would be very relevant, but due to space restrictions have to be left out of the analysis. This is also the case in terms of which narrative analysis we have chosen. Here we, in line with our psychoanalytical approach to fantasy and desire, have focused on narratives as fantasmatical stories, imaginaries that involve the gendered dimension of nationalist story-making in which past, present and future are intertwined. These fantasies are always connected to power relations, which we specifically spell out in terms of their gendered dimension, while (to a lesser extent and more indirectly) also use to clarify their interrelationship to other power dynamics (caste, class, religion, etc.).
Note 12. The materiality of the body. This is a very important point which we have addressed in this revised version in terms of the extended focus on motherhood and how this run through the article. Please the response to reviewer 1, note 1.
Reviewer 3 Report
I'm attaching my review, with apologies for the delay.

Author Response
Letter to the Reviewers
We would like to thank the three anonymous reviewers for their excellent comments which we have addressed in as much detail as possible.
All the reviewers note that the empirical aim of the article is slightly inconsistent as the focus on resistance in the title and in the abstract does not give proper attention to how resistance is counter-acted by Hindu nationalists. They also note that the special issue is on the pandemic and resistance and would like clarification as to what role the film Water and Varanasi play for the general argument of the special issue. These are all very valid points which we have addressed in relation to the reviewers’ comments.
Reviewer 3.
Reviewer 3 brings up a number of very important observations, which we believe we have responded to through our responses to Reviewer 1 and 2 as discussed below.
Note 1. The two cases and the special issue. Please see the response to Reviewer 1, note 1 and 2.
Note 2. The article does not deal with resistance and when it does deal with resistance, the analysis is not consistent and there is no analytical distinction between the themes in the analysis of Water. Together with changing the title and the focus, we have changed and reduced this section significantly to account for how Hindutva is enforced in the face of resistance. Please see response to Reviewer 1, note 1 and 2 and to Reviewer 2, note 3 and 4.
Note 3. The focus is in both cases now on how protests/resistance has been countered by Hindutva fantasies of motherhood, and how such fantasies have provided an ideological foundation for ontological security seeking practices of Vedic science as pseudoscience.
Note 4. By making a clearer connection between narrative fantasies of motherhood as Bharat Mata, anti-national protests are defined as violating such fantasies, which justifies the return to Vedic science as pseudoscience. The reviewer is right to point out that Modi himself has attempted to keep a distance from some of these pseudoscientific fantasies, and that he increased his popularity later on in the pandemic, which we now point out in the manuscript. However, at heart of these fantasies of motherhood lies a nationalism that is ultimately exclusionary and which rests on ontological security seeking practices that brand all alternative versions of Bharat Mata as anti-national.
Note 5. We have inserted some further information about the fieldwork and how the two cases have been approached through the interviews and articles. The theoretical discussion is however important to the analysis, even if space constraints does not allow us to fully go into the details of these debates. However, we believe the combination of the theoretical discussion of fantasies, gender and ontological security is well illustrated in the revised version.
Round 2
Reviewer 2 Report
This is a much-improved version of the manuscript, which is much more convincing than the previous version.
The aim and the focus of the study are much clearer, and the use of the theoretical approach is also clearer. The data is much better presented and motivated, which is good. The narrative approach is also more clearly presented, and the comparative element is toned down. The narratives are now used as examples in the argument.
In relation to the special issue, is the focus on resistance and Covid much improved as well.